# Sargachromenol Purified from *Sargassum horneri* Inhibits Inflammatory Responses via Activation of Nrf2/HO-1 Signaling in LPS-Stimulated Macrophages

**DOI:** 10.3390/md19090497

**Published:** 2021-08-31

**Authors:** Eui-Jeong Han, Thilina U. Jayawardena, Jae-Hyuk Jang, Ilekuttige Priyan Shanura Fernando, Youngheun Jee, You-Jin Jeon, Dae-Sung Lee, Jeong-Min Lee, Mi-Jin Yim, Lei Wang, Hyun-Soo Kim, Ginnae Ahn

**Affiliations:** 1Research Center for Healthcare and Biomedical Engineering, Chonnam National University, Yeosu 59626, Korea; iosu5772@naver.com; 2Department of Food Technology and Nutrition, Chonnam National University, Yeosu 59626, Korea; 3Department of Marine Life Science, Jeju National University, Jeju 63243, Korea; tuduwaka@jejunu.ac.kr (T.U.J.); youjinj@jejunu.ac.kr (Y.-J.J.); 4Anticancer Agent Research Center, Korea Research Institute of Bioscience and Biotechnology, Cheongju 28116, Korea; jangjh@kribb.re.kr; 5Department of Marine Bio-Food Sciences, Chonnam National University, Yeosu 59626, Korea; shanurabru@jnu.ac.kr; 6Department of Veterinary Medicine, Veterinary Medical Research Institute, Jeju National University, Jeju 63243, Korea; yhjee@jejunu.ac.kr; 7Interdisciplinary Graduate Program in Advanced Convergence Technology & Science, Jeju National University, Jeju 63243, Korea; 8National Marine Biodiversity Institute of Korea, 75, Jangsan-ro 101 gil, Janghang-eup, Seocheon 33662, Korea; daesung@mabik.re.kr (D.-S.L.); lshjm@mabik.re.kr (J.-M.L.); mjyim@mabik.re.kr (M.-J.Y.); 9College of Food Science and Engineering, Ocean University of China, Qingdao 266003, China; leiwang2021@ouc.edu.cn

**Keywords:** *Sargassum horneri*, sargachromenol, anti-inflammatory effect, Nrf2/HO-1 signaling, RAW 264.7 macrophages

## Abstract

In this study, we isolated sargachromenol (SC) from *Sargassum horneri* and evaluated its anti-inflammatory effect in lipopolysaccharide (LPS)-stimulated RAW 264.7 macrophages. SC did not show cytotoxicity at all concentrations and effectively increased the cell viability by reducing the nitric oxide (NO) and intracellular reactive oxygen species (ROS) production in LPS-stimulated RAW 264.7 macrophages. In addition, SC decreased the mRNA expression levels of inflammatory cytokines (IL-1β, IL-6, and TNF-α) and inflammatory mediators (iNOS and COX-2). Moreover, SC suppressed the activation of nuclear factor kappa-light-chain-enhancer of activated B cells (NFκB) and mitogen-activated protein kinase (MAPK) signaling, whereas activated the nuclear factor erythroid 2-related factor 2/heme oxygenase-1 (Nrf2/HO-1) signaling in LPS-stimulated RAW 264.7 macrophages. Interestingly, the anti-inflammatory effect of SC was abolished by the inhibition of HO-1 in LPS-stimulated RAW 264.7 macrophages. According to the results, this study suggests that the antioxidant capacity of SC leads to its anti-inflammatory effect and it potentially may be utilized in the nutraceutical and pharmaceutical sectors.

## 1. Introduction

Seaweeds are currently a popular and common food source around the world as they are rich in nutrients [1]. Seaweeds are reported to contain various substances, including acetogenins [2], oxylipins [3], polyphenols [4], terpenoids [5], steroids [6], and nucleosides [7]. The active compounds in seaweeds have been studied for various biological activities, such as antioxidant [8], anti-cancer [9], anti-diabetic [10], anti-obesity [11], anti-inflammatory [12], and anti-hypertensive effects [13]. *Sargassum* species, which are brown algae (Phaeophyceae) known to contain large amounts of polysaccharides, chromanols, and pigments, are known for their superior anti-inflammatory, anti-cancer, and antioxidant effects [9,14,15]. In particular, *Sargassum horneri* (*S. horneri*), an edible alga, is an annual species that grows along the subtidal areas of China, Japan, and Korea and has been studied for its antioxidant, anti-inflammatory, anti-tumor, and UV-protective effects [5,16]. However, most studies have focused on the biological capacities of the polysaccharides and apo-9 fucoxanthinone derived from *S. horneri* [5,17]. Hence, it is important to analyze the new organic compounds found in *S. horneri* and functionally characterize their biological effects.

The immune response plays a crucial role in preventing infections, but an overactive immune response, such as chronic inflammation, can cause various diseases, including cancer, inflammatory arthritis, obesity, migraines, dermatitis, atherosclerosis, coronary artery diseases, and insulin resistance [18]. Especially, the overactivation of macrophages results in the generation of inflammatory mediators, such as nitric oxide (NO), inducible nitric oxide syntheses (iNOS), cyclooxygenase (COX)-2, and prostaglandins, through the activation of the nuclear factor-kB (NFκB) and mitogen-activated protein kinase (MAPK) pathways, resulting in inflammatory diseases [17,19]. Also, the activated macrophages increase the generation of intracellular ROS as well as the production of NO followed by inflammatory responses [20,21]. So, downregulation of intracellular ROS generation is an important target to inhibit inflammation responses [22].

In this study, we first reveal the anti-inflammatory effect of sargachromenol (SC), a terpenoid compound derived from *S. horneri* by downregulating the intracellular ROS and NO generations in LPS-stimulated RAW 264.7 macrophages.

## 2. Results

### 2.1. SC Decreased NO and ROS Production in LPS-Stimulated RAW 264.7 Macrophages without Cytotoxicity

Before evaluating the anti-inflammatory activity of SC, we carefully examined the cytotoxicity of SC in the RAW 264.7 macrophages. As shown in Figure 1a, various concentrations of SC (7.8–62.5 µg/mL) did not significantly impair the cell viability or the cell morphology of the RAW 264.7 macrophages. We also evaluated the effect of SC on the viability of the LPS-stimulated RAW 264.7 macrophages. As shown in Figure 1b, LPS-stimulation significantly decreased the cell viability (80.23 ± 1.55%) compared to the control cells. However, various doses of SC increased the cell viability decreased by the LPS stimulation, similar to the control cells (98.00 ± 0.32%). Next, to determine whether SC modulates the production of inflammatory mediators, we analyzed the effect of SC on NO and ROS production in the LPS-treated RAW 264.7 macrophages. The results showed that SC dose-dependently inhibited LPS-induced NO production (Figure 1c). SC also effectively decreased the intracellular ROS production (Figure 1d). Thus, we showed that SC inhibited LPS-induced inflammation by reducing the NO and ROS production in RAW 264.7 macrophages.

### 2.2. SC Downregulated the Expression Levels of Inflammatory Cytokines, iNOS and COX-2 Proteins in LPS-Stimulated RAW 264.7 Macrophages

We next sought to verify the molecules that mediate LPS-induced inflammation, and the effect of SC on these molecules using reverse transcription polymerase chain reaction (RT-PCR) and Western blot analysis. First, we checked the mRNA expression levels of pro-inflammatory cytokines using RT-PCR. As shown in Figure 2a, the expression levels of IL-1β, IL-6, and TNF-α were significantly increased by LPS stimulation. However, pre-treatment with SC markedly reduced the mRNA expression level of these cytokines. The results showed that LPS stimulation induced the upregulation of iNOS and COX-2 expression, whereas they were downregulated by the pre-treatment of SC in the RAW 264.7 macrophages (Figure 2b). Taken together, these results indicate that SC has the anti-inflammatory effect by downregulating these inflammatory mediators.

### 2.3. SC Inhibited the Activation of the MAPK and NFκB Signaling Pathways in LPS-Stimulated RAW 264.7 Macrophages

We examined the inhibitory effect of SC on activation of the MAPK and NFκB signaling pathway in LPS-stimulated RAW 264.7 macrophages using Western blot analysis. As shown in Figure 3a, SC suppressed the LPS-induced phosphorylation of p38, extracellular signal regulated kinase (ERK), and c-jun N-terminal kinase (JNK). Next, to determine whether SC affects the downstream signaling of MAPK, we checked the effect of SC on NFκB signaling pathway. As shown in Figure 3b, SC suppressed LPS-induced phosphorylation of p65 and degranulation of IκBα in the cytosol. Moreover, SC inhibited the translocation of cytoplasmic p65 into the nucleus in LPS-stimulated RAW 264.7 macrophages (Figure 3c).

### 2.4. SC Mitigates LPS-Induced Inflammation via Modulating the Nrf2/HO-1 Signaling Pathway

Next, we confirmed the effect of SC on the Nrf2/HO-1 signaling pathway known as an antioxidant system. As shown in Figure 4, LPS stimulation decreased the expression of HO-1 and Nrf2, whereas co-treatment with SC significantly increased their expression in RAW2 64.7 macrophages. These results suggest that SC effectively decreased the inflammatory response by activating the Nrf2/HO-1 signaling pathway.

### 2.5. Inhibition of HO-1 Reduces the Anti-Inflammatory Effect of SC in LPS-Stimulated RAW 264.7 Macrophages

To confirm that SC elicits its effects on cell viability and NO and ROS production through the upregulation of HO-1 expression, we treated the cells with zinc protoporphyrin IX (ZnPP), a HO-1 inhibitor in the absence or presence of SC. As shown in Figure 5a–c, co-treatment with SC lead to the anti-inflammatory effect with the increased cell viability and the decreased NO and intracellular ROS production. Interestingly, the anti-inflammatory effect of SC was abolished in the presence of ZnPP, indicating that SC exerts its anti-inflammatory effects through the upregulation of HO-1 expression. Finally, we examined the activation of the NFκB pathway following ZnPP treatment in the absence or the presence of SC. As shown in Figure 5d, SC significantly inhibited the activation of NFκB signaling, whereas it was abolished the blockage of HO-1 following co-treatment with ZnPP suppressed the ability of SC. Thus, these results indicated that HO-1 is an indispensable and important mediator in the anti-inflammatory effect of SC.

## 3. Discussion

Recently, many researchers have reported several chromanol compounds of the terpenoids family, pigments, polyphenols, and sterols derived from *Sargassum* species and their anti-oxidant and anti-inflammatory effects [14,23,24,25,26]. Especially, recent studies have suggested the anti-inflammatory activities of the polysaccharides and loliolide derived from *S. horneri*, whereas there is no report on the biological effect of SC [16,17]. Therefore, it is important to evaluate the anti-inflammatory effects of SC derived from *S. horneri*.

Among the various physiological processes, the chronic inflammatory response is an important process that can lead to various chronic diseases. Factors associated with inflammatory responses include NO, PGE_2_, and inflammatory cytokines, which are regulated by a variety of inflammatory modulators [27]. NO, the representative inflammatory mediator, acts as a trigger causing ROS production that lead to cell damage and cell death [28]. Recently studies have demonstrated that excessive production of NO caused oxidative damages as well as inflammatory response in cells [28]. Thus, the decrease of NO and ROS production reduces the extent of inflammatory diseases, accompanied by antioxidant effects [21]. As shown in Figure 1, SC isolated from *S. horneri* was found to effectively reduce NO and ROS production without any cytotoxicity in RAW 264.7 macrophages. This result supported the findings that *S. horneri* effectively reduced the NO and ROS production in previous studies [14,15,17]. iNOS and COX-2 are key inflammatory mediators that influence the production of NO and PGE_2_ [27]. According to the data shown in Figure 2, SC effectively reduced the expression of COX-2 protein, which induces PGE_2_ production, and the expression of iNOS protein, which induces NO production. Thus, the extracts and compounds that reduce the expression of iNOS and COX-2 are likely to be very effective in ameliorating inflammation.

The activated macrophages cause inflammatory responses with the excessive production/expression of inflammatory cytokines, such as IL-1β, IL-6, and TNF-α, followed by tissue injury and multi-organ failure [17,23]. Thus, inhibitors of theses cytokines have been considered as candidates for anti-inflammatory materials. The present study showed that SC treatment effectively leads to the anti-inflammatory effect by reducing the expression levels of inflammatory cytokines IL-1β, IL-6, and TNF-α in LPS-stimulated RAW 264.7 macrophages.

The role of NFκB and MAPK pathways in mediating SC’s anti-inflammatory effects have been studied extensively [21,27,29]. Generally, NFκB nuclear transcription factors include the p50 and p65 proteins, which activate the inflammatory signaling pathway by the phosphorylation of the IκB complex [21]. Free forms of p50 and p65 are translocated to the nucleus, leading to stimulation of the transcription of pro-inflammatory modulators [30]. Therefore, NFκB signaling plays a very important role in anti-inflammatory mechanisms. Furthermore, MAPKs, known as upstream molecules of NFκB, mediate signal transduction from the cell surface to the nucleus in response to external stimuli and regulate cellular activity by mediating gene expression, mitosis, differentiation, survival, and apoptosis [31]. MAPKs consist of three kinases: p38, JNK, and ERK. Previous studies have shown that kinases of the MAPK pathways are activated by LPS [32]. Since phosphorylation of MAPKs causes inflammation, inhibition of the expression of phosphorylated p38, JNK, and ERK causes anti-inflammatory effects mediated through the MAPK pathways [32]. Therefore, we identified the inhibitory effect of SC on the activation of NFκB and MAPK in LPS-stimulated RAW 264.7 macrophages. Our results showed SC inhibited the activation of NFκB signaling by downregulating the phosphorylation of cytosolic IκBα and p65 as well as the translocation of p65 into the nucleus. These results indicated that SC effectively decreased the inflammation by downregulating the activation of NFκB and MAPK pathways in LPS-stimulated RAW 264.7 macrophages (Figure 3).

Under normal conditions, Nrf2 binds to the control protein Kelch-like ECH-associated protein 1 (Keap1), which promotes the degradation of Nrf2 [33]. LPS stimulation increases the expression of Keap1, thereby reducing the cytosolic levels of Nrf2, which then translocate to the nucleus, attaches to an antioxidant response element, and leads to the transcriptional activation of HO-1, a cell-protective protein [33,34]. We discovered that SC effectively activated the Nrf2/HO-1 pathway in LPS-stimulated RAW 264.7 macrophages (Figure 4). Interestingly, as presented in Figure 5, the inhibition of HO-1 abolished the anti-inflammatory effect of SC in LPS-stimulated RAW 264 macrophages. Therefore, we suggest that HO-1 is an indispensable and important mediator in the anti-inflammatory effect of SC.

## 4. Materials and Methods

### 4.1. Materials

The murine macrophage cell line RAW 264.7 was purchased from the Korean cell line bank (KCLB, Seoul, Korea). Dulbecco’s modified Eagle’s medium (DMEM), penicillin-streptomycin, and fetal bovine serum (FBS) were purchased from Gibco BRL (Burlington, ON, Canada). Dimethyl sulfoxide (DMSO) and 3-[4,5-dimethylthiazole-2-yl]-2,5-diphenyltetrazolium bromide (MTT) were purchased from Sigma Chemical Co. (St. Louis, MO, USA). The cDNA synthesis kit was purchased from Promega Co. (Fitchburg, WI, USA). TRIzol reagent was purchased from Molecular Research Center (Cincinnati, OH, USA). Primary antibodies against iNOS, COX-2, p65, phospho-p65, IκBα, phospho-IκBα, JNK, phospho-JNK, ERK, phospho-ERK, p38, phospho-p38, HO-1, and Nrf2 were purchased from Cell Signaling Technology (Beverly, MA, USA). The other chemical reagents used were of the highest grade available commercially. Ethanol extract from *S. horneri* (SHE) was prepared using the method described in our previous studies [31,35,36]. SC purified from SHE was kindly provided by the sample library of the Natural Medicine Research Center, Korea Research Institute of Bioscience and Biotechnology (KRIBB, Chungcheongbuk-do, Korea). Briefly, air-dried *S. horneri* were ground to a fine powder and extracted with 70% ethanol to obtain the crude extract, SHE. SHE was dissolved in distilled water and partitioned according to the polarity using n-hexane and ethyl acetate, respectively. The solvent fractions were concentrated using a rotary evaporator. Further, bioactive hexane fraction was fractionated using an ODS open column chromatography by hexane/ethyl acetate step-gradient elution. The best fraction was subjected to further purification using Prep HPLC using a semipreparative C18 (Cosmosil, 10 µm, 10 × 250 mm) column on a YL900 HPLC system (Young Lin, UK). The ESI-MS data for SC are shown in the Appendix A.

### 4.2. Cell Culture

The RAW 264.7 macrophages were maintained in a humidified atmosphere with 5% CO_2_ at 37 °C in Dulbecco’s modified Eagle’s medium (DMEM) supplemented with 10% FBS and 1% penicillin/streptomycin, and sub-cultured every two days.

### 4.3. Analysis of Cell Viability

Effect of SC on the cell viability was identified using an MTT assay. RAW 264.7 macrophages (2 × 10^4^ cells/well) were treated with the various concentrations of SC (7.8–62.5 µg/mL). After 24 h, 15 µL of MTT stock solution (5 mg/mL) was added to the cells for 4 h. The stained formazan crystals in the cells were dissolved with 150 µL of DMSO reagent, and the absorbance was measured at 570 nm using a SpectraMax M2 microplate reader (Molecular Devices, San Jose, CA, USA). In addition, to evaluate the effects of SC on LPS-induced cytotoxicity in RAW 264.7 macrophages, the cells were incubated with non-cytotoxic doses of SC (7.8–62.5 µg/mL) at 37 °C for 1 h. Additionally, LPS (1 µg/mL) was added to the cells, and after 24 h, an MTT assay was performed.

### 4.4. Measurement of NO Production

To evaluate the effect of SC on NO production in LPS-treated RAW 264.7 macrophages, we performed an NO assay according to the Griess assay method. The cells (2 × 10^4^ cells/well) were pretreated with various concentrations of SC (7.8–62.5 µg/mL) for 1 h, and then incubated for 24 h with LPS (1 µg/mL). After the incubation, 100 µL of culture supernatant was mixed with an equal volume of Griess reagent (1% sulfanilamide and 0.1% N-[naphthyl] ethylenediamine dihydrochloride in 2.5% H_3_PO_4_) at room temperature (20–22 °C) for 10 min. The absorbance was measured at 540 nm using a microplate reader.

### 4.5. Measurement of Intracellular ROS Production

A DCF-DA assay was performed to examine the effect of SC on intracellular ROS production. The cells (2.5 × 10^4^ cells/well) were pretreated with different doses of SC (15.6–62.5 µg/mL) prior to LPS (1 µg/mL) stimulation. After 1 h, the cells were treated with the DCFH-DA reagent (500 µg/mL) and the florescence was measured by recording the absorbance values at an excitation wavelength of 485 nm and an emission wavelength of 525 nm using a microplate reader.

### 4.6. Analysis of mRNA Expression of Inflammatory Cytokines

The cells were pretreated with SC (15.6–62.5 µg/mL) for 1 h prior to stimulation with 1 µg/mL of LPS for 12 h, and then, total RNA was isolated from the cells using TRIzol reagent. cDNA was synthesized from the RNA using a cDNA synthesis kit (TaKaRa Bio Inc., Otsu, Japan) according to the manufacturer’s protocol. PCR was carried out for 35 cycles using the corresponding primers for IL-1β, IL-6, and TNF-α and the PCR conditions used were 5 min of denaturation at 94 °C, 1 min of annealing at 55–60 °C, and 20 min of extension at 72 °C in a TaKaRa PCR Thermal Cycler (TaKaRa Bio Inc., Otsu, Japan). The PCR products were electrophoresed on 1.5% ethidium bromide/agarose gels and visualized under UV transillumination (Vilber Lourmat, Marne la Vallee, France).

### 4.7. Analysis of iNOS and COX-2 Protein Expression Levels

Western blot analysis was performed to investigate the effect of SC on the protein levels of inflammation-associated molecules such as iNOS and COX-2. The cells were incubated with various doses of SC, and cytoplasmic protein was extracted using RIPPA buffer (Thermo Fisher Scientific, Rockford, IL, USA). The proteins (40 g) were subjected to sodium dodecyl sulfate polyacrylamide gel electrophoresis (SDS-PAGE) on 12% gels and transferred to polyvinylidene difluoride membranes. The membranes were separately incubated with primary antibodies against iNOS (1:1000, Cell Signaling Technology, Danvers, MA, USA), COX-2 (1:1000), and β-actin (1:3000) and the secondary antibodies: the horseradish peroxidase (HRP)-conjugated anti-mouse or anti-rabbit IgG (1:5000). The bands were detected using enhanced Super Signal West Femto Maximum Sensitivity Substrate (Thermo Fisher Scientific, Burlington, ON, Canada) and analyzed using NIH Image J software (Version No.1, US National Institutes of Health, Bethesda, MD, USA).

### 4.8. Analysis of Activation of MAPK and NFκB Signaling Pathway 

Next, we examined the effect of SC on the protein expression in the MAPK and NFκB pathways. Cytoplasmic and nuclear proteins from cells were extracted using protein extraction kits (Thermo Fisher Scientific, Rockford, IL, USA) and the obtained proteins (40 μg) were used for the Western blot analysis. The primary antibodies used were antibodies against IκBα (1:1000, Cell Signaling Technology Inc.), p-IκBα (1:1000), p65 (1:1000), p-p65 (1:1000), p-p38 (1:1000), p38 (1:1000), p-ERK (1:1000), ERK (1:1000), JNK (1:1000), p-JNK (1:1000), and β-actin (1:3000); the secondary antibodies were HRP-conjugated anti-mouse and anti-rabbit IgG (1:5000). The subsequent steps were identical to those described earlier.

### 4.9. Effect of SC on Cell Viability, NO and ROS Production after Inhibiting HO-1 in RAW 264.7 Macrophages

To determine whether HO-1 affects the protective effects of SC against LPS-induced inflammation, we analyzed cell viability, NO production, and intracellular ROS production following SC treatment after inhibiting HO-1 expression in RAW 264.7 macrophage. The cells were incubated with 5 μM ZnPP in the presence or absence of SC before LPS (1 µg/mL) stimulation. Then, MTT, NO, and DCF-DA assays were performed as described in the previous sections.

### 4.10. Effect of SC on Activation of the NFκB Pathway after Inhibiting HO-1 in RAW 264.7 Macrophages

To investigate the effect of SC on the activation of the NFκB pathway in RAW 264.7 macrophage following the inhibition of HO-1, we performed Western blot analysis. Cells were incubated with 5 μM ZnPP for 1 h, with or without SC before LPS stimulation. The Western blot assay was performed as described in the earlier sections.

### 4.11. Statistical Analysis

Data were analyzed using the SPSS package (Version 21, IBM, Armonk, NY, USA). Values are expressed as the mean ± standard error (SE). The mean values of the tail intensity from each treatment were compared using one-way analysis of variance followed by Duncan’s multiple range test. A *p* value < 0.05 was considered statistically significant.

## 5. Conclusions

In conclusion, this study suggests that SC isolated from *S. horneri* has the anti-inflammatory effect and may be used as natural anti-inflammatory agent in pharmaceuticals, functional foods, and cosmeceuticals.

## Figures and Tables

**Figure 1 marinedrugs-19-00497-f001:**
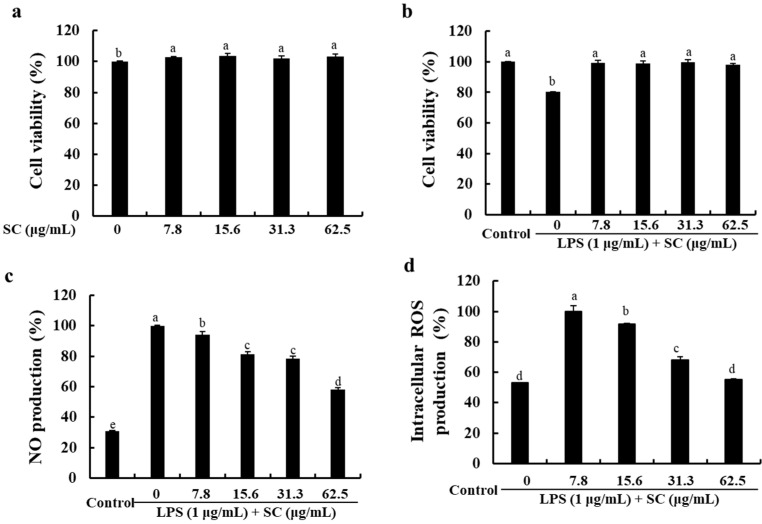
Effect of SC on cell viability and inflammatory mediators. Effect on (**a**,**b**) cell viability, (**c**) NO production, and (**d**) intracellular ROS generation in LPS-stimulated RAW 264.7 macrophages. The reproducibility of the results was confirmed by performing the experiments in triplicate (n = 3). The bars with different letters represent significant differences (*p* < 0.05).

**Figure 2 marinedrugs-19-00497-f002:**
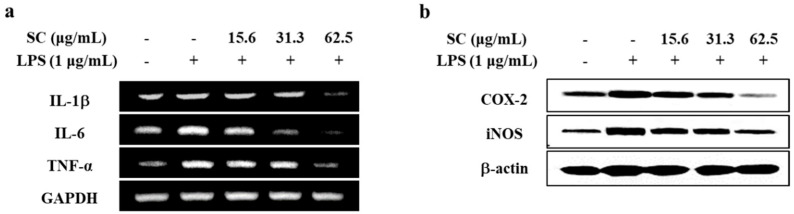
Effect of SC on the expression levels of inflammatory mediators in LPS-stimulated RAW 264.7 macrophages. Effect of SC on (**a**) mRNA expression of inflammatory cytokines (IL-1β, IL-6, and TNF-α), and (**b**) protein levels of iNOS and COX-2 in LPS-stimulated RAW 264.7 macrophages.

**Figure 3 marinedrugs-19-00497-f003:**
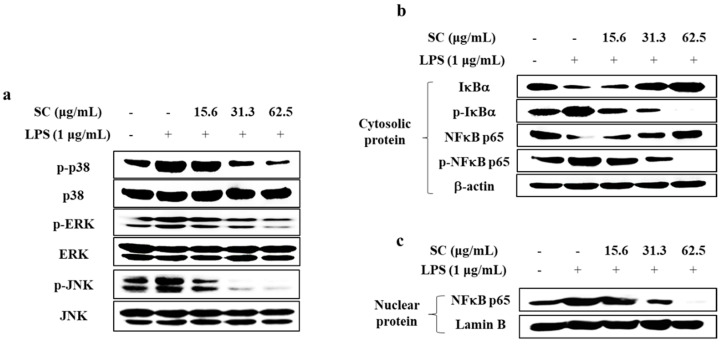
Effect of SC on activation of the MAPK and NFκB signaling pathways in LPS-stimulated RAW 264.7 macrophages. Effect of SC on (**a**) the protein expression levels of MAPK-related molecules, and (**b**,**c**) NFκB-related molecules in LPS-stimulated RAW 264.7 macrophages.

**Figure 4 marinedrugs-19-00497-f004:**
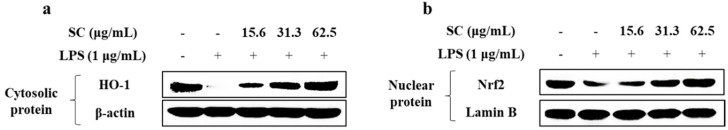
Effect of SC in the regulation of the Nrf2/HO-1 signaling pathway in LPS-stimulated RAW 264.7 macrophages. The expression levels of HO-1 in the cytosol (**a**), and Nrf2 in the nucleus (**b**).

**Figure 5 marinedrugs-19-00497-f005:**
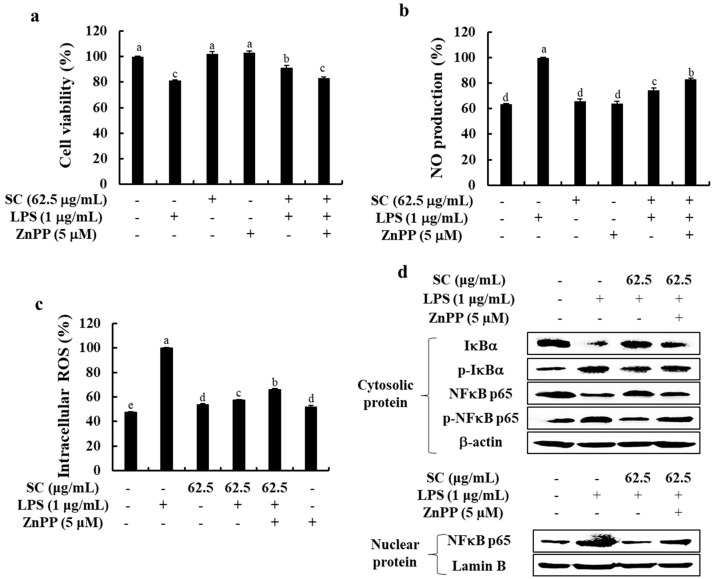
Role of HO-1 in the anti-inflammatory effect of SC on (**a**) the viability, (**b**) NO production, (**c**) intracellular ROS generation, and (**d**) the NFκB signaling pathway in LPS-stimulated RAW 264.7 macrophages. The data represent the mean ± SE of three independent experiments (*n* = 3). ^a–d^ Bars with different letters represent significant differences (*p* < 0.05).

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
