# Peer review of "Sargachromenol Purified from Sargassum horneri Inhibits Inflammatory Responses via Activation of Nrf2/HO-1 Signaling in LPS-Stimulated Macrophages"

_marinedrugs, 2021, doi:10.3390/md19090497_

Round 1

Reviewer 1 Report

The manuscript (ID: marinedrugs-1364265) entitled Sargachromenol purified from Sargassum horneri inhibits inflammatory responses via activation of Nrf2/HO-1 signaling in LPS-stimulated macrophages submitted by the authors Eui Jeong Han, Thilina. U. Jayawardena, Jae-Hyuk Jang, Ilekuttige Priyan Shanura Fernando, Youngheun Jee, You-Jin Jeon, Dae-Sung Lee, Jeong Min Lee, Mi-Jin Yim, Lei Wang, Hyun-Soo Kim Ginnaande Ahn is a solid piece of scientific work and suited for publication in the Journal Marine Drugs. However a number of instructive publications are not cited and discussed:

E. Yang et al. (2013) Sargachromenol from Sargassum micracanthum inhibits the lipopolysaccharide-induced production of inflammatory mediators in RAW 264.7 macrophages. The Scientific World Journal Volume 2013, Article ID 712303, 6 pages.

http://dx.doi.org/10.1155/2013/71230.

S. Kim et al. (2014) Anti-inflammatory effects of sargachromenol-rich ethanolic extract of Myagropsis myagroides on lipopolysaccharide-stimulated BV-2 cells. BMC Complementary and Alternative Medicine 14:231. http://www.biomedcentral.com/1472-6882/14/2313

Luo J-F, Shen X-Y, Lio CK, Dai Y, Cheng C-S, Liu J-X, Yao Y-D, Yu Y, Xie Y, Luo P, Yao X-S, Liu Z-Q and Zhou H (2018) Activation of Nrf2/HO-1 Pathway by Nardochinoid C Inhibits Inflammation and Oxidative Stress in Lipopolysaccharide-Stimulated Macrophages. Front. Pharmacol. 9:911. doi: 10.3389/fphar.2018.00911.

S. Saraswati et al. (2019) Sargassum seaweed as a source of anti-inflammatory substances and the potential insight of the tropical species: A review. Mar. Drugs, 17, 590; doi:10.3390/md17100590

S. Saraswati et al. (2021) Screening of in-vitro anti-inflammatory and antioxidant activity of Sargassum ilicifolium crude lipid extracts from different coastal areas in Indonesia. Mar. Drugs, 19, 252. https://doi.org/10.3390/md19050252

Ko, W.; Lee, H.; Kim, N.; Jo, H.G.; Woo, E.-R.; Lee, K.; Han, Y.S.; Park, S.R.; Ahn, G.; Cheong, S.H.; et al. (2021) The anti-oxidative and anti-neuroinflammatory effects of Sargassum horneri by heme oxygenase-1 induction in BV2 and HT22 cells. Antioxidants 2021, 10, 859. 

https://doi.org/10.3390/ antiox10060859

Such a discussion is essential to finally improve the submitted draft.

Author Response

We have uploaded a reply to the review as an attached file

Reviewer 2 Report

marinedrugs-1364265-peer-review-v1

The manuscript entitled "Sargachromenol Purified from Sargassum Horneri Inhibits Inflammatory Responses via Activation of Nrf2/HO-1 Signaling in LPS-stimulated Macrophages" addresses a topic relevant and suitable for publication in Marine Drugs. The manuscript is well structured and well founded. Some corrections are necessary however, as I suggest below. 

Corrections needed:

Title: Authors must follow the rules of taxonomy, and even in the title, write "Sargassum horneri" and not "Sargassum Horneri" 

line 48 - Sargassum species, which are brown algae (Phaeophyceae)

line 49/50 -  pigmented compounds (= pigments)

line 235 -  Materials and Methods - Since sargachromenol is the component tested in this experimental work, I suggest that the authors do not limit themselves to citing a bibliographic source for the extraction methodology used. Since this compound was extracted by the same team that wrote this manuscript, I am of the opinion that you should include a short paragraph summarizing its extraction methodology. 

Author Response

(The authors gave the same response as above.)
